# Modeling CRISPR-Cas13d on-target and off-target effects using machine learning approaches

Xiaolong Cheng [1,2,6], Zexu Li [3,6], Ruocheng Shan[1,4], Zihan Li [3], Shengnan Wang[3], Wenchang Zhao[3], Han Zhang[3], Lumen Chao[1,2], Jian Peng [5], Teng Fei [3] ✉ & Wei Li [1,2] ✉

A major challenge in the application of the CRISPR-Cas13d system is to accurately predict its guide-dependent on-target and off-target effect. Here, we perform CRISPR-Cas13d proliferation screens and design a deep learning model, named DeepCas13, to predict the on-target activity from guide sequences and secondary structures. DeepCas13 outperforms existing methods to predict the efficiency of guides targeting both protein-coding and non-coding RNAs. Guides targeting non-essential genes display off-target viability effects, which are closely related to their on-target efficiencies. Choosing proper negative control guides during normalization mitigates the associated false positives in proliferation screens. We apply DeepCas13 to the guides targeting lncRNAs, and identify lncRNAs that affect cell viability and proliferation in multiple cell lines. The higher prediction accuracy of DeepCas13 over existing methods is extensively confirmed via a secondary CRISPR-Cas13d screen and quantitative RT-PCR experiments. DeepCas13 is freely accessible via http://deepcas13.weililab.org.

CRISPR-Cas13, including Cas13a, Cas13b, Cas13c, Cas13d (RfxCas13d, or CasRx) and the newly discovered Cas13X/Y[1], belongs to the type VI CRISPR-Cas system that exclusively targets single-stranded RNA (ssRNA)[2–4]. All Cas13 nucleases contain two HEPN domains as RNase to cleave RNAs or to process precursor crRNAs into mature crRNAs. Once activated by the single guide RNAs (sgRNAs) bearing complementarity sequences to the target RNA, Cas13 will cleave the target RNA and also nearby RNA molecules. Cas13 nucleases have also been engineered into efficient, multiplexable, and specific tools for the knockdown, editing and recognition of RNAs (and methylated RNAs like m[6]A RNA) in mammalian cells[5–7]. In addition, Cas13 has been a central component of several rapid and sensitive methods for detecting viral infections including SARS-COV-2[8–11].

A major challenge in the application of the CRISPR-Cas system (including Cas13) is to design sgRNAs with high on-target efficiency and specificity. On the one hand, an accurate prediction of sgRNA efficiency would facilitate the optimized design of sgRNAs with maximized on-target efficiency (i.e., high sensitivity). On the other hand, understanding the specificity of Cas nucleases will help to avoid the potential off-target effects, possibly due to the off-target cleavage at the DNA (for Cas9) or RNA (for Cas13) level, respectively, or due to its unwanted collateral cleavage to nearby mRNA molecules (Cas13) in many applications. For this reason, CRISPR screening has been a cost-

[1]Center for Genetic Medicine Research, Children's National Hospital, Washington, DC 20010, USA. [2]Department of Genomics and Precision Medicine, George Washington University, Washington, DC 20010, USA. [3]National Frontiers Science Center for Industrial Intelligence and Systems Optimization, Key Laboratory of Bioresource Research and Development of Liaoning Province, College of Life and Health Sciences, Northeastern University, Shenyang 110819, China. [4]Department of Computer Science, George Washington University, Washington, DC 20052, USA. [5]Department of Computer Science, University of Illinois at Urbana-Champaign, Urbana, IL 61801, USA. [6]These authors contributed equally: Xiaolong Cheng, Zexu Li. ✉e-mail: feiteng@mail.neu.edu.cn; wli2@childrensnational.org

effective approach to systematically investigate the efficiency and specificity of CRISPR-Cas9/Cas13, by examining the behaviors of a large number of guides in one single experiment. Guides in screens, including proliferation/viability screens and FACS-sorting screens, have been used to investigate factors that affect knockout efficiency[12], design guides that maximize activity and minimize off-target effects[13], and to train machine learning[14] and deep learning models[15–17] for a precise prediction of guide behaviors. Similar with Cas9 screens, Cas13d screens has been used to study the specificity and efficiency of Cas13d system[18], by examining a large number of tiling guides that span the gene of interest. Unlike CRISPR-Cas9 on-target activity prediction tools[17,19–21] that only extract the spatial features of the sequence, models for Cas13 need to consider the secondary RNA structures of guides, which is a major factor for knockdown efficiency[18]. However, there are some limitations to existing methods built for Cas13d efficiency prediction. First, the training dataset is based on FACS-sorting screens that measured the expression level of a few specific genes, and it is unclear whether the corresponding model applies to guides targeting other genes and measuring other phenotypes (e.g., cell proliferation). Second, it is unclear whether such model, trained on guides targeting protein-coding genes, works on non-coding RNAs. Third, a systematic, experimental validation is lacking to evaluate the performances of existing models. Last but not least, the off-target effect of Cas13d, mostly due to its collateral cleavage to mRNAs within the cell, is not fully explored.

Understanding the off-target effect of gene editing tools (e.g., TALEN, RNAi, Cas9, base editor), either dependent or independent from the guide, has been an important and challenging task in genome engineering. On the one hand, many assays have been developed to investigate the off-target editing outcomes of Cas9 guides[22–26], although most of these techniques are limited to report the effect of one sgRNA and none of them can be directly applied to Cas13 system. On the other hand, the non-specific toxicity effect has been examined on almost every gene editing tool. For example, the overexpression of short hairpin RNA (shRNA) damages cell viability by disrupting the miRNA processing in host cells[27]. DNA double-strand breaks induced by Cas9 may also trigger DNA damage response (and subsequent cell death), particularly when guides target the amplified regions in the genome that results in stronger DNA damage[28–30]. By examining thousands of Cas9 sgRNAs targeting non-functional, non-genic regions in the genome, the non-specific toxicity of Cas9 can be identified and mitigated in a screening manner[31,32]. The collateral RNA cleavage has been one of the major sources for Cas13 off-target effect[5–7], although such effect, especially on cell viability, has not been investigated systematically.

Here we systematically model the on-target efficiency and off-target viability effect of Cas13d (CasRx), using machine learning and deep learning approaches that are trained on large-scale Cas13 screening datasets. We first conduct CRISPR-Cas13d screens that contains 10,830 guides targeting essential/non-essential genes and long non-coding RNAs (lncRNAs). Combining this dataset with published studies, we obtain data from 22,599 Cas13d sgRNAs to systematically investigate the efficiency and specificity of Cas13d. We next design DeepCas13, a deep learning-based model for predicting CRISPR-Cas13d on-target activity. DeepCas13 takes advantage of convolutional neural network and recurrent neural network to learn spatial-temporal features from the sequences and secondary structures of sgRNAs. DeepCas13 outperforms traditional machine learning methods and previous published tools, and demonstrates good performance in predicting guides targeting non-coding RNAs (e.g., circular RNAs and long non-coding RNAs). In addition, we systematically evaluate the off-target viability effect of Cas13d, by investigating guides targeting non-essential genes in the viability screens. We find that features determining the off-target viability effect of a guide are very similar to features associated with on-target efficiency. Such

effect can be mitigated in the Cas13d screens, using guides that target non-essential genes as negative controls. Compared with using non-targeting guides as negative controls, this approach greatly reduces false positives in the screen, a finding that resembles the use of non-essential guides in CRISPR/Cas9 screens. We apply these on-target and off-target models to CRISPR-Cas13d viability screens that target 234 lncRNAs, and identify putative lncRNAs whose perturbation reduces cell fitness in different cell lines. Finally, we design and perform a secondary validation screen, followed by quantitative RT-PCR (qRT-PCR) experiments, to confirm the higher accuracy of DeepCas13 predictions over existing methods. DeepCas13 is freely accessible via a web server at http://deepcas13.weililab.org/.

## Results

### Cas13d proliferation screen and a convolutional recurrent neural network for efficiency prediction

To systematically investigate the efficiency and specificity of Cas13d, we conducted a two-vector CRISPR/Cas13d proliferation screening experiment (Fig. 1a; see "Methods" for details). The screening library contains 10,830 sgRNAs targeting 192 protein-coding genes and 234 lncRNAs, and the screening experiment was performed using a melanoma cell line A375. The library targets 94 known essential genes and 14 non-essential genes, which are identified from previous RNA interference and CRISPR screens[33,34] and are confirmed to be essential (or non-essential) in A375, respectively (Supplementary Fig. 1a), providing a unique dataset (3934 guides in total and about 30 guides per gene) to model Cas13d sgRNA efficiencies. In addition, this dataset may potentially overcome the biases of previous tiling screening datasets, where guides that only target 2–3 genes are used. At the end of the screening, the abundance of Cas13d sgRNAs was evaluated using high-throughput sequencing, and the data analysis was performed using the MAGeCK algorithm we previously developed[35]. Overall, the quality of the screen is high based on multiple quality control (QC) measurements (Supplementary Table 1), including sequencing depth (around 5.6 million reads per sample), the average number of reads per guide (over 300), the number of missing guides (less than 4), and the low Gini index values (less than 0.06) indicating the non-biased distribution of guides across different conditions. In addition, 20 out of 94 known essential genes are significantly depleted (FDR < 10%; Supplementary Fig. 1b, Supplementary Data 1). These genes include MTOR (in mTOR pathway), TUBA1B (component of Tubulin complex), RPL4/RPS8 (ribosomal subunit), all known to be involved in essential functionalities of cellular functions. The guides targeting essential genes are strongly depleted, as expected, demonstrating the success of the screens (Supplementary Fig. 1c).

Combining with published Cas13d screening datasets (Supplementary Table 2), we obtained 22,599 guides that target both coding genes and non-coding elements, a unique dataset to train and evaluate predictive models for Cas13d. The large number of guides (10,279 guides targeting essential genes in proliferation screens or marker genes in FACS screens; see Supplementary Data 1) enabled us to design DeepCas13, a deep learning model to predict Cas13d sgRNA on-target efficiencies (Fig. 1b). Guides targeting non-coding elements were excluded from the training, as non-coding RNAs are usually expressed at lower levels than protein-coding genes, and their functions on cell proliferation or viability remain largely unclear. DeepCas13 uses sgRNA sequences and its predicted secondary RNA structures as input, two known features determining on-target efficiency from previous studies[36,37]. The output of DeepCas13 is a score, named "Deep Score" and ranging between 0 and 1, to indicate the sgRNA on-target efficiency. Features from sgRNA sequences and structures are extracted through convolutional recurrent neural networks (CRNN), which is commonly used to extract features in both spatial and temporal dimensions. These sgRNA spatial-temporal features are then concatenated in one layer, followed by a fully

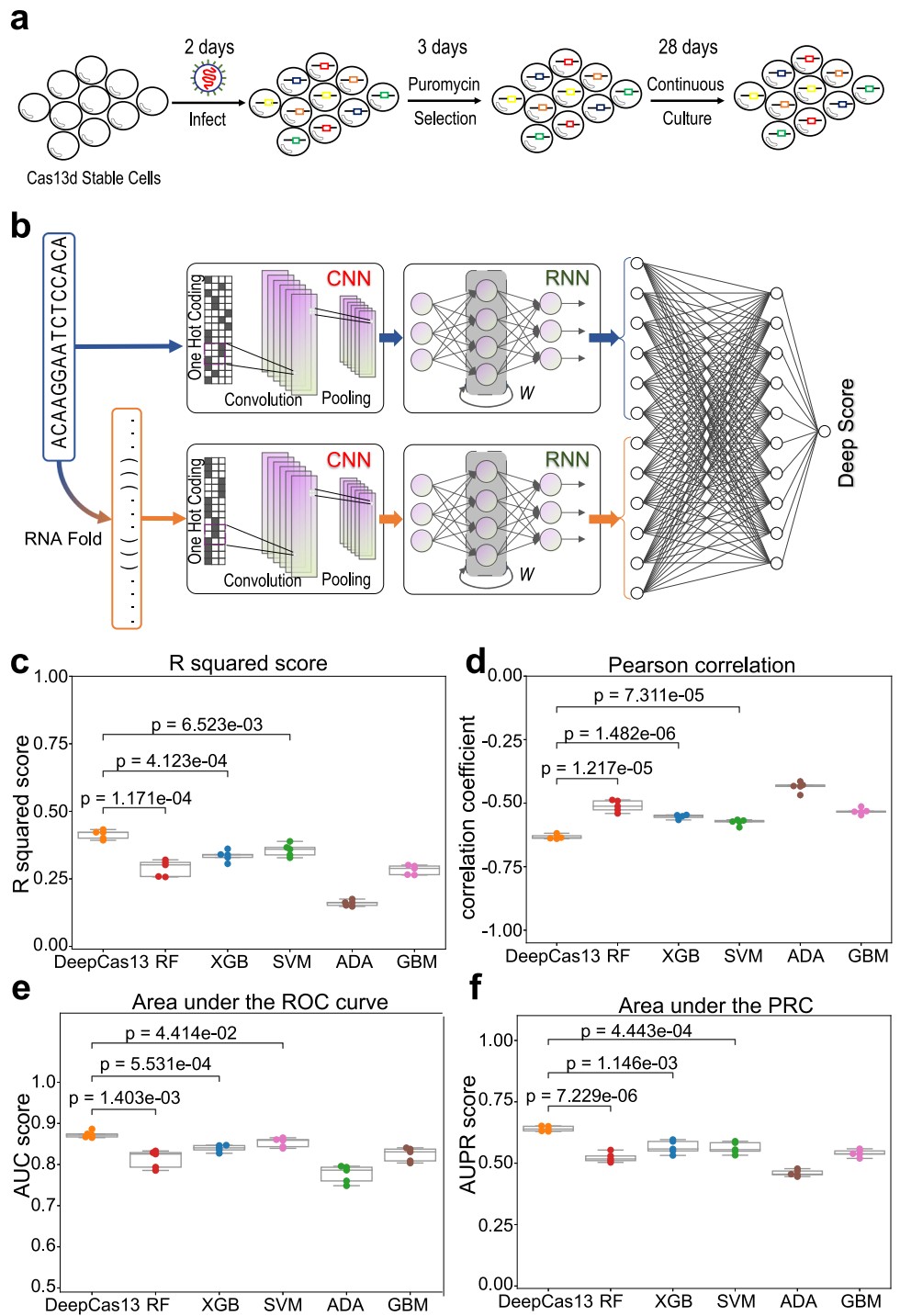

**Fig. 1 | DeepCas13 predicts Cas13d on-target activity effectively. a** A schematic view of Cas13d pooled screen experiment. The library targets both protein-coding genes and non-coding RNAs. **b** A schematic view of DeepCas13. It extracts spatial-temporal features of sgRNA sequence and RNA secondary structure using convolutional and recurrent neural networks. **c** Comparison of the predictive performance using R squared score. $n = 5$ folds used in cross-validation. The two-sided independent t-test is used for analysis. **d** The Pearson correlation coefficient between predicted scores and log2 fold change (LFC). The predicted scores are from five-fold cross-validation. $n =$ five folds used in cross-validation. The two-sided independent t-test is used for analysis. **e** AUC score comparison. Any guide with LFC $< -0.5$ is regarded as positive sample. $n = 5$ folds used in cross-validation. The two-sided independent t-test is used for analysis. **f** Area Under the Precision-Recall Curve (AUPR) scores. Any guide with LFC $< -0.5$ is regarded as positive sample. $n = 5$ folds used in cross-validation. The two-sided independent t-test is used for analysis. The top, mid-line and bottom of the boxplot (**c–f**) represents the upper quartile (Q3), median, and lower quartile (Q1), respectively. The ends of the whiskers represent the minimum and maximum values in the data set.

connected layer in neural network for prediction (see "Methods" for more details).

DeepCas13 is different from machine learning models developed for CRISPR-Cas9 efficiency prediction[17,19–21], as it takes the predicted

secondary structures as input, in addition to sgRNA sequence features. Indeed, strongly depleted sgRNAs targeting essential genes usually have higher values of minimum free energy (MFE) from secondary structures (Supplementary Fig. 2a). As a result, information from the

predicted secondary structures improves the precision of the model (Supplementary Fig. 2b–g, $p = 0.00125$).

## DeepCas13 outperforms other methods for Cas13d sgRNA efficiency prediction

We compared DeepCas13 with five conventional machine learning methods, including Random Forest (RF), XGBoost (XGB), Support Vector Machine (SVM), AdaBoost (ADA), and Gradient Boosting (GBM). For conventional machine learning methods, 185 curated features as previous described[18] were manually generated for training (Supplementary Data 2). All methods were trained and tested from three published Cas13d tiling screening datasets (5,726 sgRNAs in total) using five-fold cross validation.

We evaluated the performance of different models on (1) predicting the performances of all guides in the dataset, and (2) classifying guides into efficient (or non-efficient) categories. For the first evaluation (all the guides), we compared the coefficient of determination ($R^2$) value and Pearson correlation coefficient (PCC) between the predicted scores and the actual log fold changes (LFCs) (Fig. 1c, d). For the second evaluation (classification), we split all guides into two different groups (LFC < = −0.5 as positive guides and the rest as negative guides), and calculated the area under the Receiver Operator Characteristic (ROC) curve (AUC, Fig. 1e) and area under the precision-recall curve (AUPR, Fig. 1f) for each method. DeepCas13 has a higher $R^2$ and a stronger negative PCC coefficient than other methods using 5-fold cross-validation (Fig. 1c, d), indicating the better performance of DeepCas13 over other methods. Similarly, the average AUC score (from 5-fold cross-validation) of DeepCas13 was 0.87, compared with the score ranging from 0.78 to 0.85 for other methods (Fig. 1e and Supplementary Fig. 3). The average AUPR score, which is a better metric to evaluate the performance on unbalanced dataset (i.e., fewer positive samples), is 0.64 for DeepCas13, which is significantly higher than other approaches (ranging between 0.45 and 0.58; Fig. 1f, Supplementary Fig. 4), demonstrating the better performance of Deep-Cas13d on classifying guides into strong or weak knockdown effects.

We next compared DeepCas13 with a recently published random forest prediction model[18], which is denoted as RF$_{NBT}$ below. We used the RF$_{NBT}$ model, which is already trained and provided by the authors, directly for the evaluation. Since RF$_{NBT}$ only uses FACS screening datasets for training, we first evaluated whether Deep-Cas13, trained on FACS and additional proliferation screening datasets, outperformed RF$_{NBT}$. For one of the two proliferation screens, denoted as A375$_{NBT}$, we used a five-fold cross validation strategy to only use 80% of the guides (plus all guides in FACS screens) for training, and used the remaining 20% guides for performance evaluation. This process was repeated five times, and the ROC scores, Pearson correlation coefficients and precision-recall scores were recorded for both proliferation screens (Fig. 2a–c and d–f for the screens in previous study[18] and in this study, respectively). Deep-Cas13 better distinguishes efficient and inefficient sgRNAs than RF$_{NBT}$ (Fig. 2a), demonstrated by its higher AUC scores (0.84 and 0.74 for DeepCas13 and RF$_{NBT}$, respectively). In addition, there was a much stronger correction between the predicted Deep Scores and LFC distribution (Fig. 2b, PCC = −0.46 and −0.28 for DeepCas13 and RF$_{NBT}$, respectively). For the precision and recall values (Fig. 2c), most of the sgRNAs with high predicted scores from DeepCas13 were indeed efficient (i.e., high precision), and the average AUPR score for DeepCas13 is 0.5049, in comparison with 0.2774 for RF$_{NBT}$ (Fig. 2c). Similar trends were found when another proliferation screening dataset (screening in Fig. 1a; denoted as A375$_{DeepCas13}$) was used for training and evaluation (Fig. 2d–f). Collectively, both DeepCas13 and RF$_{NBT}$ performed well at ranking sgRNAs, with most negative cases at one end of a scale and positive cases at the other. Compared with RF$_{NBT}$, DeepCas13 identified fewer false positive cases, indicated by its higher precision value than RF$_{NBT}$ with the same recall value.

We further evaluated the performance of DeepCas13 and RF$_{NBT}$ by using one of the proliferation screening datasets as an independent validation dataset ("leave-one-dataset-out"; Supplementary Fig. 5). DeepCas13 reaches higher AUC and AUPR values than RF$_{NBT}$ (Supplementary Fig. 5), although the advantage of DeepCas13 was not as strong as in Fig. 2. There was no significant difference in AUC or AUPR values when proliferation dataset was removed from training (Supplementary Fig. 5), indicating that there were few common spatial-temporal features learned between the two proliferation datasets. That may be due to the intrinsic differences within two proliferation datasets, including guide lengths (i.e., 22 bp in our study vs 27 bp in previous study[18]), whose effect may not be captured in the training dataset. Despite that, DeepCas13 consistently outperforms RF$_{NBT}$, possibly due to the deep learning framework as well as the additional guides for training from the proliferation screens (1398 guides for 35 essential genes in research[18], and 3155 sgRNAs for 94 essential genes in our study, respectively, see Supplementary Data 3).

We also applied Integrated Gradients (IG)[38], an explainable AI framework to estimate the relationship between a model's predictions in terms of its features. Based the output of IG, we calculated the preferences of sequence position and nucleotide composition in the guide sequence. As shown in Fig. 2g and Supplementary Fig. 5e, f, highly efficient guides tend to contain C/G rather than A/T in the positions 15–23 of the guide sequence, consistent with findings from another similar study[39].

## DeepCas13 predicts the efficiencies of sgRNAs targeting non-coding RNAs

Having demonstrated the performance of DeepCas13 on protein-coding genes, we next evaluated whether DeepCas13 can predict the efficiency of guides targeting non-coding RNAs, including circular RNAs (circRNAs) and long non-coding RNAs (lncRNAs). circRNA is a single-stranded non-coding RNA that forms a covalently closed, continuous loop from non-canonical splicing event[40]. circRNAs have been implicated in human physiology and diseases, although their functions are largely unclear due to a lack of adequate methods to study them[41]. Recently, CRISPR-Cas13d has been successfully applied to study the functions of circRNAs in a screening manner[42,43], providing a novel yet efficient approach to systematically investigate circRNA functions.

We first applied DeepCas13 to one circRNA screening dataset[42] where >2500 human hepatocellular carcinoma-related circRNAs were screened. DeepCas13 successfully distinguished efficient sgRNAs from inefficient sgRNAs (Fig. 3a, AUC score is 0.7492), although the area under the precision-recall curve (AUPR) was quite low for all the guides (Fig. 3b). Since most of the circRNAs are considered to be nonfunctional, the low AUPR score may be because of the majority of the guides that have little (or no) change in abundance, even if they are predicted as efficient. Therefore, we only focused on guides that target top negative selected circRNAs (identified from the MAGeCK algorithm), and found the AUPR score greatly increased, up to 0.61 for the top 10% negatively selected circRNAs (Fig. 3b).

We also applied DeepCas13 to another circRNA screening study[43], where 3800 sgRNAs were designed to target sequences across back-splicing junction (BSJ) sites of highly expressed human circRNAs in three different cell lines separately. Similarly, we limited our prediction to guides that target top 50 negatively selected circRNAs, and classified guides whose log fold change smaller than −0.5 as efficient guides. Overall, DeepCas13 predicts these circRNAs with high precision and recall (Fig. 3c, AUPR scores are 0.8037, 0.5680, and 0.6821 for HEK293FT, HT29, and Hela cells, respectively). On the other hand, the AUC scores of all guides in different cell lines vary (Fig. 3d), ranging from 0.63 (HEK293FT cells) to 0.54 (HT29 cells), possibly indicating some cell-type-specific effect that cannot be captured using sequence and RNA secondary structures. As a result, for those sgRNAs with high Deep Scores, many are actually not depleted (false positives), possibly

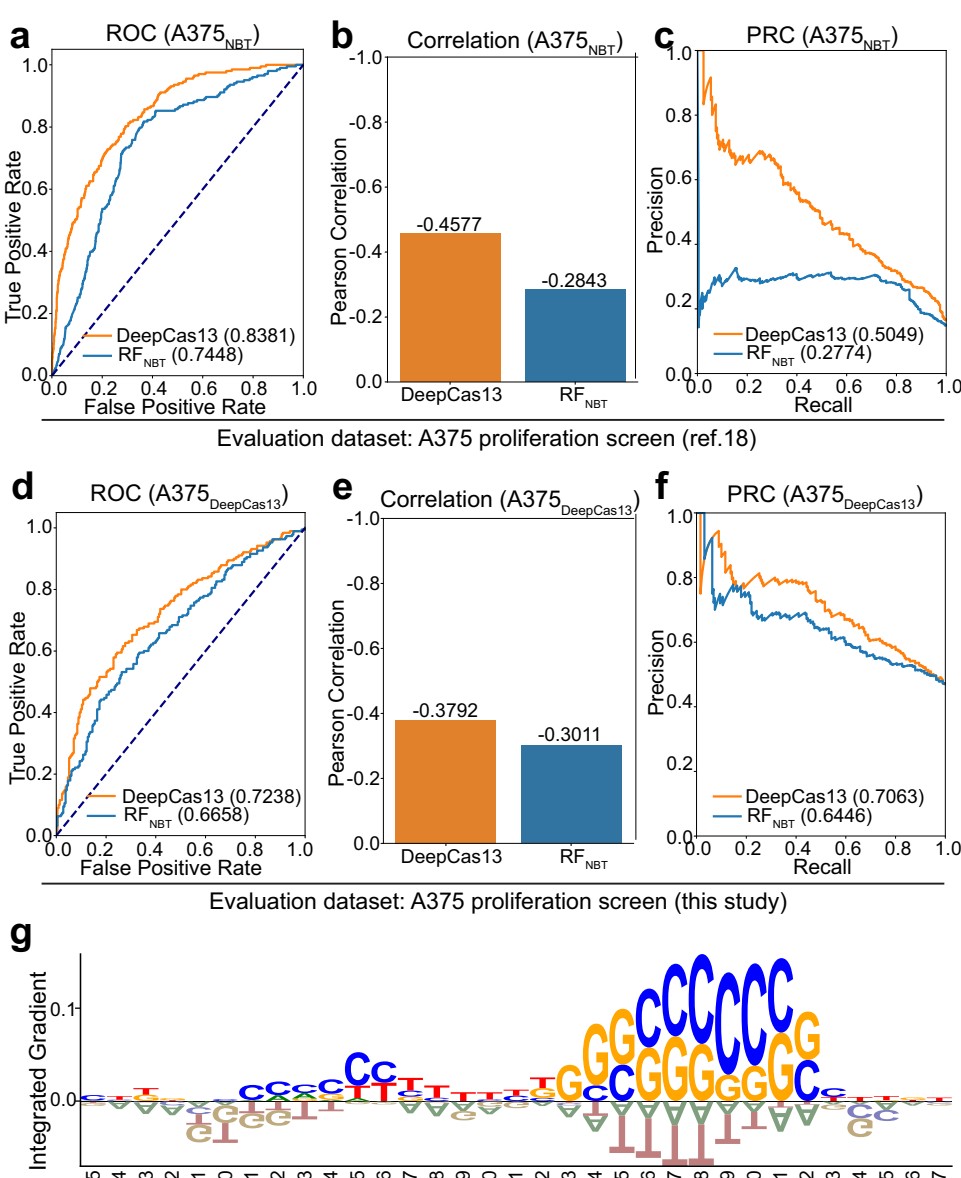

**Fig. 2 | Performance comparison of DeepCas13 with state-of-the-art tool. a** ROC curve comparison for the public Cas13d proliferation dataset. **b** Correlation comparison for the public Cas13d proliferation dataset. **c** PRC curve comparison for the public Cas13d proliferation dataset. **d** ROC curve comparison for our Cas13d proliferation dataset. **e** Correlation comparison for our Cas13d proliferation dataset. **f** PRC curve comparison for our Cas13d proliferation dataset. **g** Feature importance at each position by Integrated Gradients (IG).

because the corresponding circRNAs are not functional (Fig. 3e, AUPR scores are 0.4262, 0.2412, and 0.3344 for HEK293FT, HT29 and Hela cells, respectively).

Next, we applied DeepCas13 to guides targeting 234 lncRNAs in our Cas13d screen (Fig. 1a). Since not all the lncRNAs affect cell viability, we identified top 20 negatively selected lncRNAs using the MAGeCK algorithm[35,44], and examined their predicted Deep Scores and the log fold change values (Fig. 3f). The correlation between Deep-Cas13 scores and the actual log fold changes is strong (Pearson correlation coefficient = −0.27, $p$ value = 5.12e−5), demonstrating a good prediction power for DeepCas13 on lncRNAs. Overall, DeepCas13 demonstrated a satisfactory performance in predicting the on-target activity of sgRNAs targeting circRNAs and lncRNAs.

**Studying the off-target viability effect using machine learning approaches**

We next investigated the off-target viability effect of Cas13d, or the unintended effect of Cas13d on cell viability, by examining guides that

target known non-essential genes (derived from RNAi or CRISPR/Cas9 screens[33,34] and are confirmed as non-essential in A375; Fig. 4a; Supplementary Fig. 1a). The rationale is that, since targeting these genes are confirmed to have little (or no) effect on cell proliferation and viability, the strong depletion of guides targeting these genes should come from its effect on cell viability, possibly by cleaving unintended RNA molecules (off-target RNAs or collateral RNAs). Such non-specific toxicity has been reported in other genome editing tools including shRNA[27] and Cas9 nuclease[28–30], and can be systematically evaluated in a screen manner, by examining thousands of Cas9 sgRNAs targeting the non-functional elements in the genome[31,32].

We identified guides that have the strongest depletion in the screen (lowest LFC value), and identified features that mostly correlate with these guides. In total, 2893 guides targeting non-essential genes from two distinct proliferation datasets (one from published viability screen[18] and the other from our own Cas13d screen in Fig. 1a) are examined. We used random forest (RF) to train the model and identify feature importance, as the number of training samples is too few for

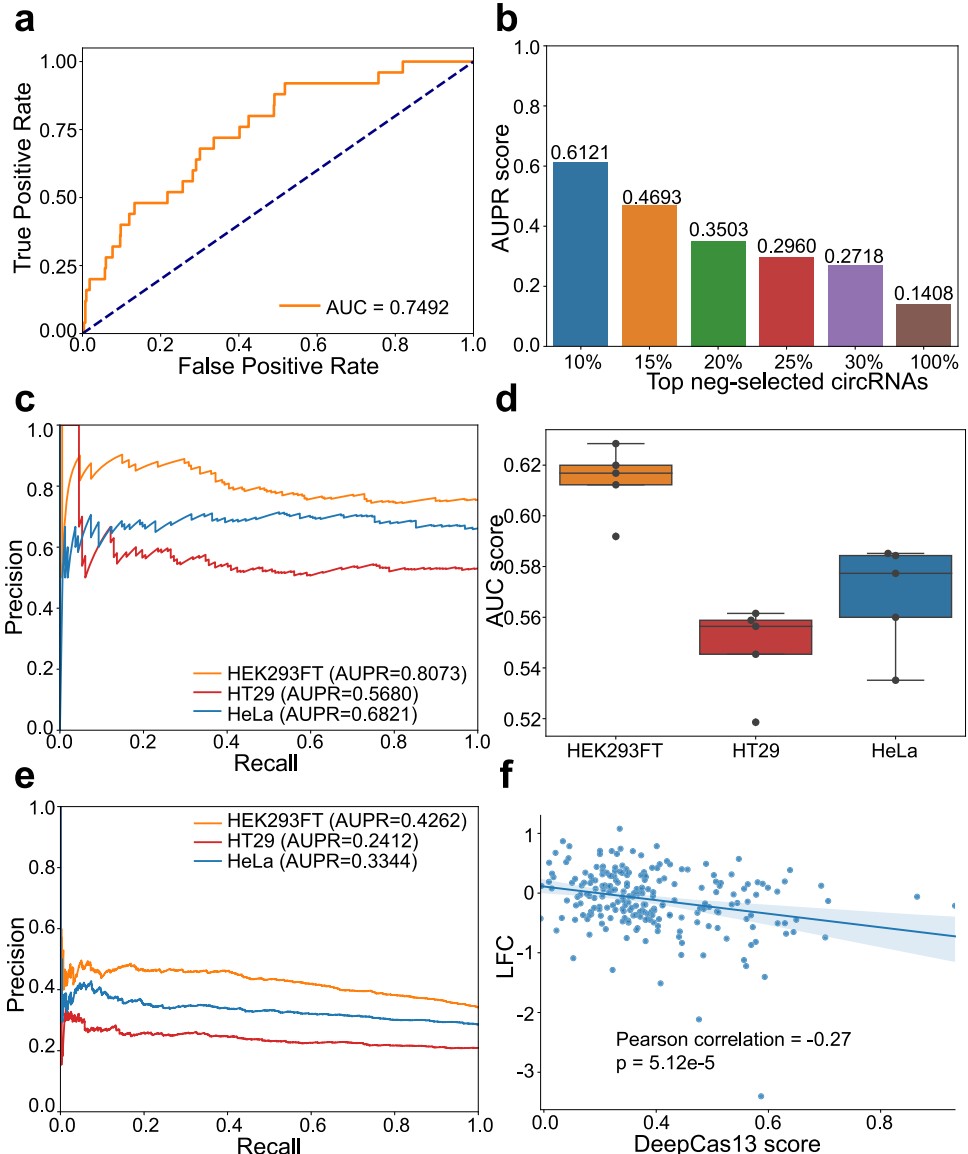

**Fig. 3 | Predict activity of sgRNAs targeting non-coding sgRNAs. a** ROC curve for circRNA screening dataset and the AUC score is shown in the legend. **b** Average precision score of the sgRNAs associated to top 10, 15, 20, 25, 30, and 100% negative selected genes. **c** PRC curves of the sgRNAs associated to top 50 negative selected genes for HEK293FT, HT29, and Hela cells. **d** AUC score distribution from five-fold cross-validation for HEK293FT, HT29 and Hela cells. $n = 5$ folds used in cross-validation. **e** PRC curves for HEK293FT, HT29, and Hela cells. The AUPR scores are shown in the legend. **f** The correlation between DeepCas13 predicted scores and the actual log2 fold changes for guides targeting top 20 negatively selected lncRNAs. The *p*-value is two-sided and calculated by a test of the null hypothesis that the distributions underlying the samples are uncorrelated and normally distributed. The top, mid-line, and bottom of the boxplot (**d**) represents the upper quartile (Q3), median, and lower quartile (Q1), respectively. The ends of the whiskers represent the minimum and maximum values in the data set.

deep learning frameworks. Interestingly, feature weights derived from these non-essential targeting guides closely resemble features determining gRNA on-target activity, which are derived from comparing guides with strong vs. weak dropouts in essential genes using random forest (Fig. 4b). These feature weights are highly consistent between the two datasets used (Fig. 4c), demonstrating that the off-target viability effect of Cas13d is closely related to the guide's on-target cleavage effect, which has been extensively studied in previous sections. Indeed, the lowest predicted energy from guide, or guides plus their direct repeat (DR) sequence ("Energy" and "crRNA energy" in Fig. 4d, respectively), two known features for predicting crRNA on-target efficiency, are ranked top in both on-target and off-target activity models (Fig. 4d). On the other hand, whether the guides are located on the coding region ("CDS" in Fig. 4d) affects its on-target efficiency, but not off-target viability effect. In addition, in three published FACS-

sorting screens that do not use viability as readout, guides that have strong on-target gene depletion ("efficient guides") also have stronger off-target viability score (trained from non-essential genes; Supplementary Fig. 6a), a demonstration that the off-target effect of guides also closely correlated with their on-target activities in other screening types.

**Controlling the off-target viability effect in Cas13d screens targeting genes and lncRNAs**
The unintended off-target viability effect of Cas13d guides is similar to Cas9 guides, where its DNA cleavage activity activates DNA damage response in the host cells and affects cell viability[30]. Such effect is also observed in the screening settings, where guides targeting amplified regions of the cells have stronger depletion effect and generates false positives in the screen[28,29,45]. For this reason, Cas9 guides targeting

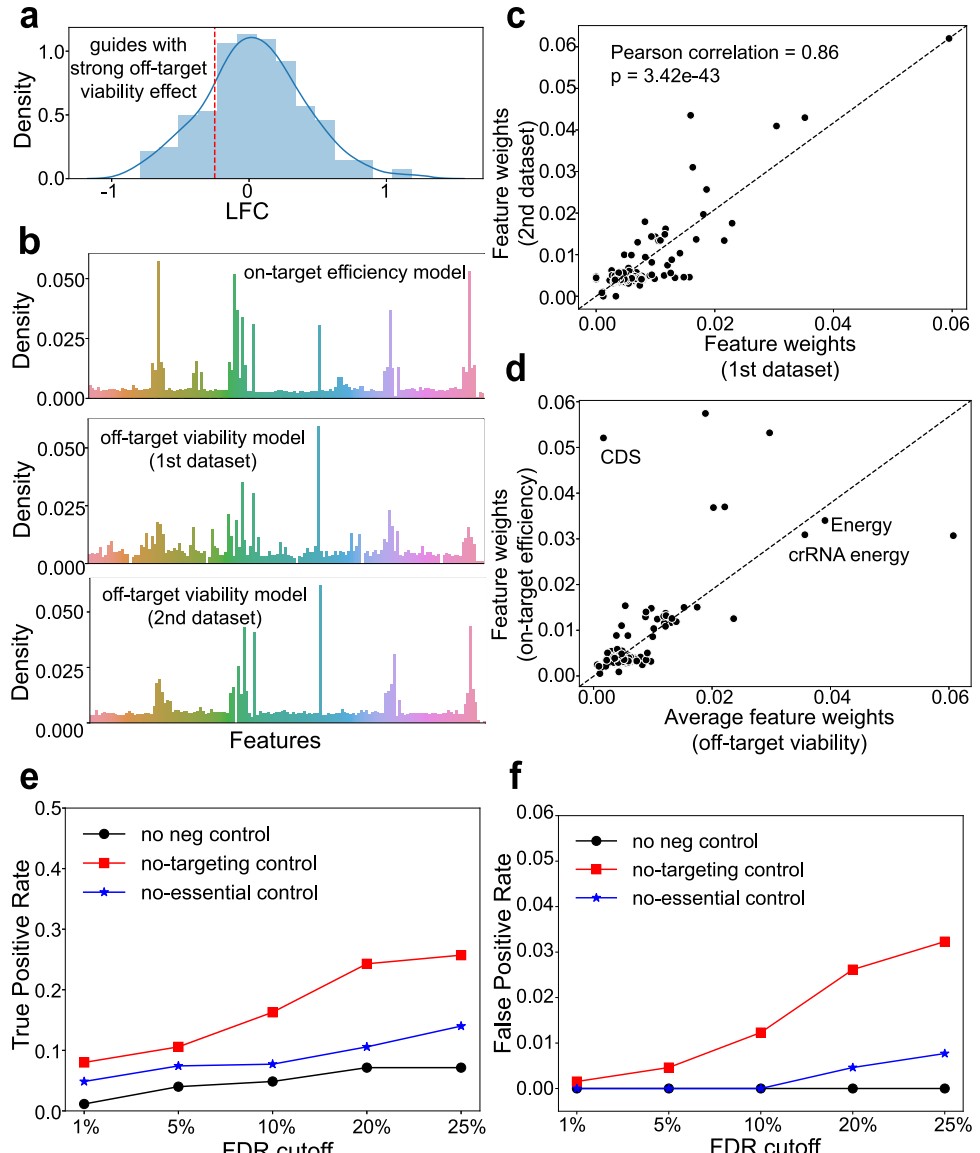

**Fig. 4 | The off-target viability effect of Cas13d guides. a** Study design. Guides that have strong depletion in targeting known non-essential genes are compared with other guides. **b** The weights of features trained by random forest by comparing guides with strong vs. weak depletion in essential genes ("on-target model", top) and non-essential genes ("off-target viability model", middle and bottom. **c** The Pearson correlation of feature weights across two different datasets. The *p*-value is two-sided and calculated by a test of the null hypothesis that the distributions underlying the samples are uncorrelated and normally distributed. PCC: Pearson Correlation Coefficient. **d** Comparing feature weights across on-target and off-target model. **e** The true positive rate (in identifying known essential genes) with different FDR cutoff using different controls: no controls, non-targeting controls or non-essential controls. **f** The false positive rate (in identifying non-essential genes as significant) as in (**e**). In (**e**, **f**), the A375 cell proliferation screens from research[18] was used.

non-essential genes or non-coding regions are recommended as a better negative control[31–33]. To compare the choice of negative control guides in Cas13d screens (non-essential genes vs non-targeting guides vs no negative control guides used), we examined the true positive rate (in identifying known essential genes as significant) and false positive rate (in identifying known non-essential genes as significant), respectively, using different FDR cutoffs (Fig. 4e, f) in one Cas13d proliferation screening dataset[18]. We observed low false positive rate (but also low true positive rate) without using any negative controls. Similar with Cas9 screens, using non-targeting guides as controls has the highest true positive rate but also introduces a large number of false positives in the screen. In contrast, using guides targeting non-essential genes demonstrated reasonable true positive rate (Fig. 4e) and also a much lower false positive rate compared with non-targeting controls (Fig. 4f). Similar trends were also found on another

proliferation screening dataset (Supplementary Fig. 6b, c). These results demonstrated that, guides targeting non-essential genes should be used instead of non-targeting guides in Cas13d screens to reach a good balance in true positive and false positive rate.

We applied our findings to the screening data that targets both protein-coding genes and lncRNAs (Fig. 5a). Comparing with default setting (not using any controls), using non-essential guides as negative controls increases the number of essential genes as well as lncRNAs that are statistically significant (Fig. 5a). In A375 cells, 20 lncRNAs are identified as negatively selected with statistically significance (FDR < 0.25), including those that are known to be associated with tumorigenesis (Fig. 5b). For example, *NEAT1* has been shown to promote the proliferation, migration and invasion of melanoma cells as well as A375 cells, by interfering the regulation of multiple microRNAs and their target genes[46,47]. Another example is *SNHG29*, whose tumorigenesis

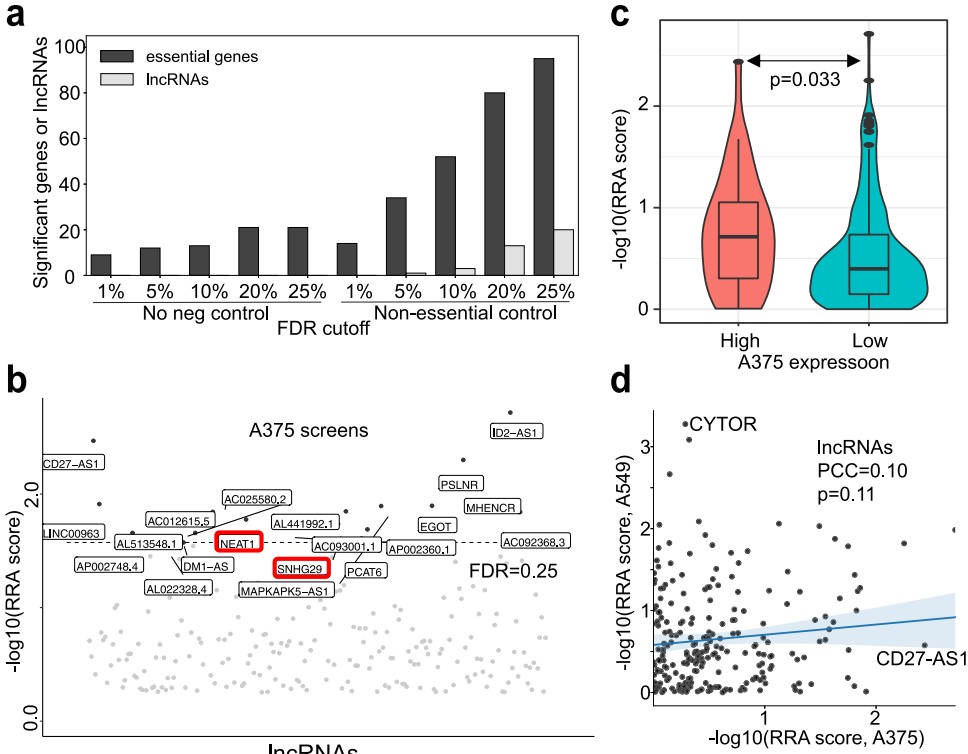

**Fig. 5 | Screens for genes and lncRNAs. a** The number of statistically significant genes and lncRNAs using different FDR cutoff, using either no negative control guides or using guides targeting non-essential genes ("non-essential control") as negative controls. **b** Top negatively selected lncRNAs, identified from the MAGeCK algorithm. **c** The distribution of Robust Rank Aggregation (RRA) scores, measured in the screen, of lncRNAs with high (or low) expressions in A375 cells. $n = 89$ lncRNAs. The two-sided independent t-test is used for analysis. **d** The RRA scores of lncRNAs across two different cell lines. Two lncRNAs that showed distinct

phenotype in two cell lines (CD27-AS1 and CYTOR) are marked. Error band shows the 95% confidence interval for the regression estimate. The *p*-value is two-sided and calculated by a test of the null hypothesis that the distributions underlying the samples are uncorrelated and normally distributed. PCC: Pearson Correlation Coefficient. The top, mid-line, and bottom of the boxplot (**c**) represents the upper quartile (Q3), median, and lower quartile (Q1), respectively. The ends of the whiskers represent the minimum and maximum values in the data set.

role has been reported in multiple cancer types[48,49]. The over-expressions of both *NEAT1* and *SNHG29* are all associated with poor survival in TCGA melanoma cohort (Supplementary Fig. 7a, b). We further performed the same screening experiment on A549 cell line (a lung cancer cell line; Supplementary Fig. 7c), and compared the lncRNA screening results with their expression levels across two different cell lines (A375 and A549; Fig. 5c, d; Supplementary Fig. 7d). lncRNAs that have higher expressions are more likely to be negatively selected (Fig. 5c). In addition, their functions show a stronger cell type-specific effect than protein-coding genes (Fig. 5d and Supplementary Fig. 7e, f), consistent with previous findings that these lncRNAs may likely work in a cell type-specific manner than protein-coding genes[50].

**Experimental validation of DeepCas13**

To experimentally benchmark the performances of DeepCas13 and RF_NBT, we designed and performed a secondary validation screen experiment in A375 cells by targeting essential genes and noncoding RNAs with Cas13d and using cell viability as screening readout (Fig. 6a). The screening library contains 12,000 sgRNAs, including 9500 sgRNAs targeting 59 essential protein-coding genes (mRNA). In addition, 38 lncRNAs and 40 circRNAs with putatively essential roles on cell growth are selected based on the results of our primary screens and published studies[42,43]. Included in the library are also 2500 gRNAs targeting non-essential genes and non-targeting guides as negative controls. Both DeepCas13 and RF_NBT were used to predict the efficiencies of these 9500 sgRNAs. We checked the LFC distribution of highest-scoring sgRNAs for each group using different thresholds (Fig. 6b and Supplementary Fig. 8a, b). The LFC distribution of top sgRNAs predicted

by DeepCas13 is significantly lower than those by RF_NBT in essential mRNA and lncRNA group, demonstrating the higher prediction accuracy of DeepCas13. Interestingly, although top guides predicted by DeepCas13 have overall lower LFC values than those predicted by RF_NBT in the circRNA group, the differences are not significant, and the median LFC values from either group of sgRNAs are close to zero (Fig. 6b and Supplementary Fig. 8a, b), possibly due to the cell type-specific effects of circRNAs that were only essential in other specific cell types rather than in A375 cells. Consistently, the LFC distribution of circRNA-targeting sgRNAs is indistinguishable from the negative control guides (Supplementary Fig. 8c, d).

We also compared the performances of guides that have the highest (and lowest) predicted efficiencies in both methods. For each target gene, we selected 10 sgRNAs that have the highest (marked as *H* in Fig. 6c) and lowest predicted scores (marked as *L* in Fig. 6c) for each method, respectively. Efficient sgRNAs predicted by both tools (labeled as $H_{both}$) have a much lower LFC distribution than none-fficient sgRNAs predicted by both tools (labeled as $L_{both}$), indicating the overall consistent prediction results from both tools (Fig. 6c and Supplementary Fig. 8e, f). The top efficient sgRNAs predicted by DeepCas13 only ($H_{DeepCas13}$) have a significantly smaller LFC values than those predicted by RF_NBT only ($H_{RFNBT}$, $p = 2.68e{-}10$). In addition, the higher correlation coefficients between DeepCas13 predicted scores and LFC values (Fig. 6d) indicated a better concordance of DeepCas13 predictions with experimental results than RF_NBT. Interestingly, when combining the power from both methods, it correctly predicts the essentiality of tested circRNAs ($H_{both}$ vs $L_{both}$) while either single method fails to do so (Supplementary Fig. 8f),

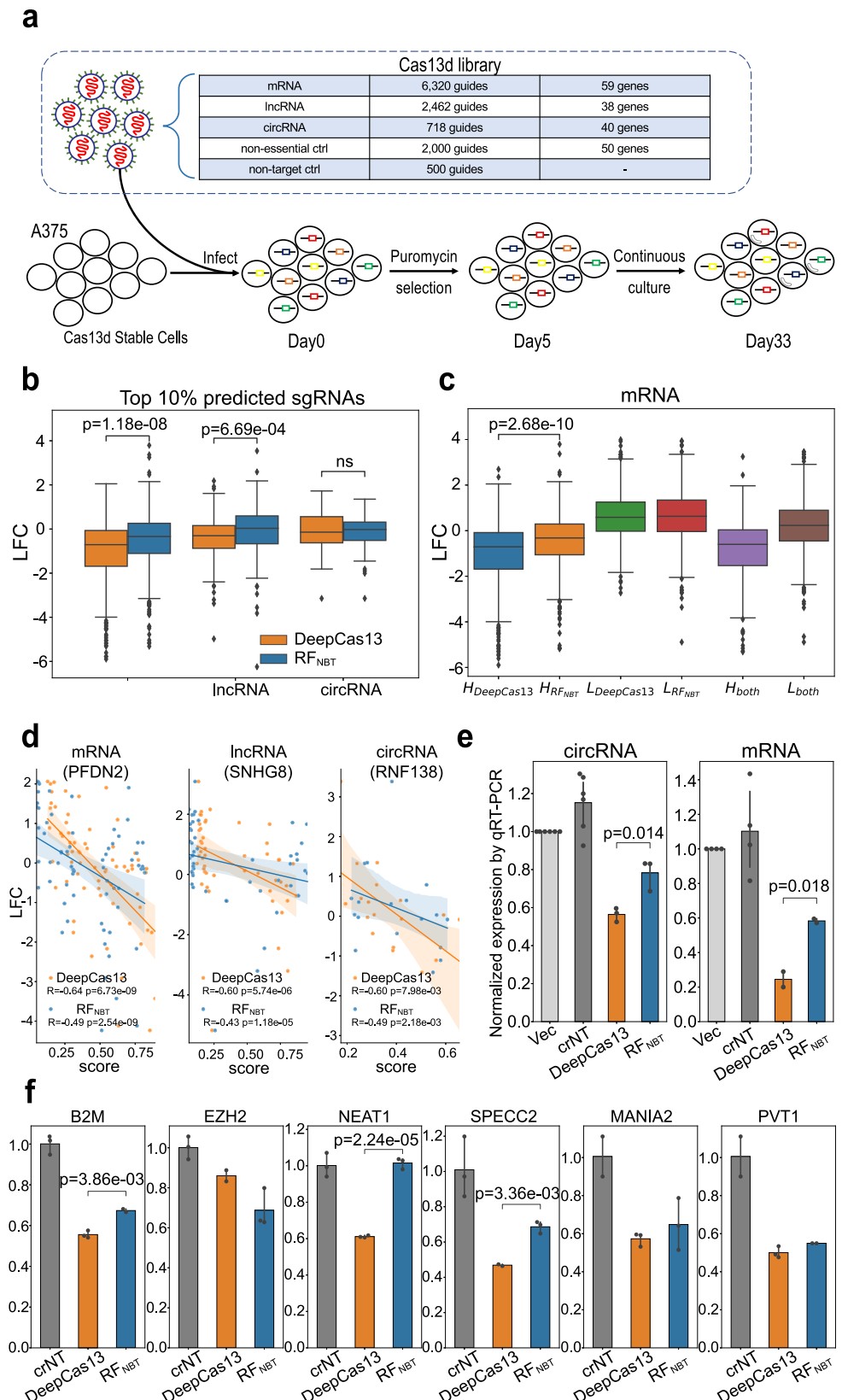

suggesting the complementary and additive functions of the two algorithms.

As an alternative validation of the screening results, we selected 10 sgRNAs with highest predicted DeepCas13 or RF$_{NBT}$ scores (5 sgRNAs for each method), and experimentally tested the knockdown efficiencies of these guides using quantitative RT-PCR (qRT-PCR). Briefly, for each sgRNA, we transiently transfected these guides into HEK293FT cells, and measured the corresponding RNA expression changes with qRT-PCR. Non-targeting guides are used as negative controls. Consistent with the screens, we found sgRNAs with highest Deep Scores better knocked down mRNA and circRNA expressions than those with highest RF$_{NBT}$ scores (Fig. 6e).

**Fig. 6 | Experimental validation of DeepCas13 performance. a** A schematic view of the validation screen experiment. The library targets essential protein-coding genes, lncRNAs, and circRNAs. **b** LFC distribution of top 10% predicted sgRNAs by each tool. $n$ = 632, 246, and 71 guides targeting mRNA, lncRNA and circRNA respectively. The two-sided independent t-test is used for analysis. **c** LFC distribution of high and low predicted sgRNAs for individual coding gene. $H$ means sgRNA with high predicted score. $n$ = 566, 557, 580, 589, 379, and 579 guides in $H_{DeepCas13}$, $H_{RFNBT}$, $L_{DeepCas13}$, $L_{RFNBT}$, $H_{both}$, $L_{both}$ group respectively. The two-sided independent t-test is used for analysis. **d** Spearman correlation between LFC and predicted scores. Error band shows the 95% confidence interval for the regression estimate. The $p$-value is two-sided and calculated by a test of the null hypothesis that the

distributions underlying the samples are uncorrelated and normally distributed. **e** qRT-PCR validation of the top predicted sgRNA from the screen experiment. crNT: non-targeting crRNA. Error bar shows the 95% Confidence Interval (CI). $n$ = 3 independent qPCR experiments. Error bar shows the 95% confidence interval. The two-sided independent t-test is used for analysis. **f** "Head-to-head" qRT-PCR validation of sgRNAs that target additional coding genes and non-coding RNAs. Error bar shows the 95% CI. $n$ = 3 independent qRT-PCR experiments. Error bar shows the 95% confidence interval. The two-sided independent t-test is used for analysis. The top, mid-line and bottom of the boxplot (**b**, **c**) represents the upper quartile (Q3), median, and lower quartile (Q1), respectively. The ends of the whiskers represent the minimum and maximum values in the data set.

In addition to the screening approach, we performed "head-to-head" comparisons between DeepCas13 and RF$_{NBT}$ using qRT-PCR. We first selected nine protein-coding genes, lncRNAs or cirRNAs that were not included in our screen, then identified two sgRNAs that have the highest predicted efficiencies by either DeepCas13 or RF$_{NBT}$ (18 sgRNAs in total). For each sgRNA, we used qRT-PCR to measure the corresponding RNA expression changes in HEK-293FT cells (Fig. 6f). Three out of the nine targets were excluded from further analysis, because activation effects, rather than inhibition effects, were consistently observed in these targets, possibly due to the secondary effects of on- and/or off- target inhibition (Supplementary Fig. 8g). In five out of the six remaining targets (Fig. 6f), DeepCas13 showed better performance than RF$_{NBT}$, including 3 statistically significant targets. In conclusion, DeepCas13 demonstrated better performance in identifying effective sgRNAs than the existing methods.

## Discussion

A better understanding (and prediction) of the efficiency and specificity of Cas13 will accelerate the application of this new RNA editing tool to many areas. In this study, we performed CRISPR-Cas13d (CasRx) viability screens that target both protein-coding genes and non-coding RNAs. Based on the screening data (and published datasets), we designed DeepCas13, a deep learning-based model for predicting CRISPR-Cas13d on-target activity. DeepCas13 uses convolutional neural network and recurrent neural network to learn the spatial-temporal features of sgRNA sequence and RNA secondary structure. Compared with five conventional machine learning methods and a recently published state-of-the-art tool, DeepCas13 demonstrated a better performance in Cas13d sgRNA efficiency prediction (Figs. 1, 2). In addition, DeepCas13 performs well on both protein-coding genes and non-coding RNAs (including circular RNAs and long non-coding RNAs; Fig. 3). The better performance of DeepCas13 over existing methods was experimentally validated using a secondary screening approach at a large scale as well as direct qRT-PCR assay on a smaller set of targeted transcripts.

We also applied machine learning method (random forest) to better understand the off-target viability effect, by comparing guides that have strong vs. weak effect on non-essential genes. We found features identified from this analysis are consistent with features for on-target efficiency, implying that such off-target viability effect is closely associated with the efficiency of a guide (Fig. 4). Therefore, a proper negative control (targeting non-essential RNAs) should be used, especially in the proliferation/viability screens, to control for such off-target viability effect. Indeed, using non-essential control guides improves the true positive rate of a screen (compared with not using any control guides), while keeping false positive rate at a lower level (compared with using non-targeting controls).

Our Cas13d screen contains guides that target 234 lncRNAs. We demonstrated that our DeepCas13 model predicts the phenotype of guides targeting top negatively selected lncRNAs (Fig. 3f), and increases the statistical power in identifying statistically significant lncRNAs using non-essential control guides (Fig. 5a). The analysis also identified known and putative oncogenic lncRNAs, as well as lncRNAs

that have cell type-specific functions between cell lines (Fig. 5b–d). Collectively, our screening system and computational model provide a simple yet effective approach for the large-scale functional studies of non-coding RNAs (including lncRNAs).

Prediction models based on machine learning and deep learning have been developed in many gene editing applications, including predicting editing efficiencies and editing outcomes from many types of enzymes like Cas9[15–17], Cas13[39], base editor[51–53], prime editor[54,55], etc. Our DeepCas13 model provides an important prediction model for Cas13. Looking forward, we expect similar deep learning frameworks to be developed to a wider range of gene editors, to harness a wide variety of research and clinical applications.

Currently, despite the datasets used in this study, few Cas13d screening datasets are available. The power of deep learning framework is therefore limited as it performs best with a large number of training datasets. In the future, new Cas13d screens can be added as additional training samples to further improve the prediction performance of DeepCas13 and our off-target viability model. Another limitation is, DeepCas13 only focused on Cas13d (CasRx), one of the commonly used Cas13 nuclease. It is therefore unclear whether DeepCas13 works on other Cas13 proteins like Cas13a, Cas13b, or Cas13XY. Once enough screening datasets (or validation datasets) for other proteins are available, DeepCas13 can be further extended to predict the efficiency of other Cas13 proteins beyond Cas13d.

## Methods
### Oligonucleotide library design
**Gene selection.** We choose 94 known "core-essential genes" and 10 known "non-essential" genes whose perturbation is known to have strong (or no) effect on cell proliferation or viability from published resources including our previous study[33,34]. In addition, 88 other protein-coding genes with various functions in cancer are added, including oncogenes (e.g., *PIK3CA*, *PAK2*, *MYC*) or tumor suppressor genes (e.g., *RB1*, *CDKN1B*).

**lncRNA selection.** We selected lncRNAs whose expressions are over-expressed in multiple cancer types in the TCGA cohort (*BRCA* and *PRAD*). Briefly, the expressions of lncRNAs for each patient sample are downloaded from TCGA, and lncRNAs are selected if they meet the following criteria: (1) their average expressions (in log2 FPKM) are greater than 2 for the 15% samples with the highest expression of this lncRNA; (2) their log2 fold changes (compared with normal samples) are greater than 0.2; and (3) their expressions (FPKM) are greater than 10 in the corresponding cancer cell lines. In addition, 25 literature-curated lncRNAs with known functions in multiple cancers are included[56].

**sgRNA design.** Ensembl gene and lncRNA annotations (version: GRCh38) are used to extract the sequences of corresponding genes or lncRNAs. For genes (or lncRNAs) with multiple transcripts, the transcripts that have the corresponding RefSeq ID are used. We first enumerate all possible guides (22 bp) that span the entire transcript, then remove guides that (1) map to more than one location in the human

genome and transcriptome (allowing up to 1 mismatch) or (2) contains BsmBI digestion sites ("CGTCTC" or "GAGACG"). The remaining guides are randomly selected if they hit the greatest number of transcripts ("common" guides) within the same gene/lncRNA. If all the guides that hit the greatest number of transcripts are selected, the remaining guides that hit the second greatest number of transcripts will be randomly selected, and so on. For essential and non-essential genes, 35 guides are designed per gene. For other genes or lncRNAs, 15 guides are designed per gene or lncRNA.

**Validation screen library design.** (1) Essential genes were ranked by their CCLE gene expression in the A375 cell line and those used in the primary library were removed from the candidate list. The top 59 high-expressed essential genes were selected in the final validation library. (2) lncRNAs expression data was downloaded from one previous study[57]. The top 38 high-expressed lncRNA was chosen as candidate targets. (3) circRNA list was downloaded from the previous Cas13d studies[42,43]. We chose the top 40 high expressed circRNAs and designed all possible sgRNAs that cross the junction. sgRNA design was in the same strategy as the primary library design.

**CRISPR library synthesis and construction**
The pooled synthesized oligos (Synbio Technologies, China) were PCR amplified and then cloned into lentiGuide-13dDR-puro vector (for expressing Cas13d sgRNA) via BsmBI site by Gibson Assembly. The ligated Gibson Assembly mix was transformed into self-prepared electrocompetent Stable *E. coli* cells by electro-transformation to reach the efficiency with at least 100× coverage representation of each clone in the designed library. The transformed bacteria were cultured directly in liquid LB medium for 16–20 h at temperature 30 °C to minimize the recombination events in *E. coli*. The library plasmids were then extracted with EndoFree Maxi Plasmid Kit (TIANGEN, Cat no. 4992194).

**Pooled genome-wide CRISPR Screen**
Human melanoma A375 cells, human non-small cell lung carcinoma A549 cells and HEK293FT cells were obtained from American Type Culture Collection (ATCC) and maintained in DMEM medium supplemented with 10% fetal bovine serum (FBS). Protein-coding gene and lncRNA-targeting plasmid libraries under lentiviral lentiGuide-13dDR-puro backbone were firstly transfected along with pCMVR8.74 and pMD2.G packaging plasmids into HEK293FT cells using Lipofectamine™ 2000 Transfection Reagent (Invitrogen, Cat no. 11668019) to generate Cas13d sgRNA-expressing lentivirus. Harvest virus-containing media at 72 h post-transfection, and spin down the media at 1000 g for 5 min to remove the floating cells and cell debris. Carefully collect the virus supernatant, aliquot and store them at −80 °C for further use. Test the virus titer and MOI (multiplicity of infection) before proceeding to the screen. For two-vector Cas13d primary cell growth screen and secondary validation screen, cas13d-expressing A375 or A549 cells were firstly generated by lentiviral infection of Cas13d (using lenti-Cas13d-NLS-blast vector) and then amplified to a number around $3 \times 10^7$. Then, these cells were infected with Cas13d sgRNA-expressing lentiviral CRISPR library with MOI -0.3. Two days later, select the infected cells with puromycin (1 μg/mL for A375 cells and A549 cells) for three days to get rid of any non-infected cells before changing back to normal media. At least ~300× coverage of cells were collected as Day 5 sample and stored at −80 °C for later genomic DNA isolation. The rest of the cells were continually cultured until Day 33 before harvesting as the end point sample. Genomic DNA from Day 5 and Day 33 samples was extracted. The regions encompassing the sgRNAs were firstly PCR-amplified for around 20–23 cycles with the following primer pair: lentiGuide-13dDR_F1: 5′- AATGGACTATCATATGCTTACCGTAACTTGA AAGTATTTCG-3′; lentiGuide-13dDR_R1: 5′- GGAGTTCAGACGTGTGCT CTTCCGATCTCCAGTACACGACATCACTTTCCCAGTTTAC-3′. The

second round of PCR were employed to attach the illumina adaptors and index for around 10–12 cycles with the following primers: library_F: 5′- AATGATACGGCGACCACCGAGATCTACACTCTTTCCCTACACGAC GCTCTTCCGATCTATCTTGTGGAAAGGACGAAACACC-3′; Index_R: 5′- CAAGCAGAAGACGGCATACGAGATNNNNNNNNGTGACTGGAGTTCA GACGTGTGCTCTTCCGATCT -3′ (N(8) are the specific index sequences). These PCR products were gel purified and pooled for high-throughput sequencing to identify sgRNA abundance on illumina PE150 sequencing platform (Novogene, China).

**Cloning of crRNA expression plasmids**
Complementary single-stranded DNA (ssDNA) oligos (each at a final concentration of 10 mM, GENEWIZ) encoding the spacer sequences were annealed with T4 DNA ligase buffer (final concentration at 1×, NEB) and 0.5 μL of T4 PNK (NEB) per 10 μL reaction. Annealing steps were as the following: 30 min at 37 °C and 5 min at 95 °C followed by a 5 °C/min ramp to 4 °C. The annealed spacer oligos (diluted 1:100) were then inserted into the crRNA backbone (lentiGuide-13dDR-puro, BsmBI digestion) via ligation with T4 DNA ligase (NEB) and 16 °C for 1 h. Spacer sequence are listed in Supplementary Data 4.

**Transient transfection of human cell lines**
For transient transfection, HEK293FT cells were plated at a density of 100,000 cells per well in a 12-well plate and transfected at >90% confluence with 1 μg of lenti-Cas13d-NLS-blast and 1 μg of crRNA expression plasmid using DNA transfection reagent (Neofect) according to the manufacturer's protocol. Transfected cells were harvested 72 h post-transfection for gene expression analysis.

**Reverse transcription and quantitative real-time PCR (qRT-PCR)**
Total RNAs from cells were extracted by the UNIQ-10 Column Trizol Total RNA Isolation Kit (Sango Biotech). cDNA was generated by the High Capacity cDNA Reverse Transcription Kit (Applied Biosystems). Quantitative RT–PCR (qRT-PCR) was performed using UltraSYBR Mixture (CWBIO) on a QuantStudio™ 5 Real-Time PCR System (Thermo Fisher). The reaction mixture was incubated at 95 °C for 10 min, and then 40 PCR cycles were performed with the following temperature profiles: 95 °C for 10 s, 60 °C for 30 s, and 72 °C for 32 s. The gene expression values were normalized to those of *RPS28* control using ΔΔCt method. qPCR primers are listed in Supplementary Data 4.

**Cas13d screening data processing**
The Cas13d screening sequence data in fastq format was processed by MAGeCK[35,44] package. The MAGeCK count command was used to generate sgRNA raw read count table. For public dataset, we used its raw read count table directly if it's provided in the original paper. Then, a Robust Rank Aggregation (RRA) algorithm was applied to normalize and rank the read counts. The sgRNA LFC was collected from the sgRNA summary table, and the top negative genes was selected from the gene summary table based on the ranking results.

**Normalize the LFC and define target value for prediction**
A customized sigmoid function was used to normalize the LFC, and the calculation formula was as follows:

$$S(x) = \frac{1}{1 + e^{-nx+b}} \tag{1}$$

where, $x$ was the LFC of a given sgRNA.
The $n$ and $b$ were two customized parameters to make:

$$\begin{cases} S(0) = 0.3 \\ S(-0.3) = 0.7 \end{cases} \tag{2}$$

These customized parameters were based on LFC distribution of the training data, and they were designed to map effective LFC (<−0.5) closer to 1 and at the same time make the mapping range of LFC enriched region ([−0.3, 0]) as large as possible.

## Development of DeepCas13

For a given Cas13d sgRNA, we first calculate its minimum free energy (MFE) structure by ViennaRNA[58]. The secondary structure is regarded as an equal contribution to the sgRNA sequence. In DeepCas13 model, the sgRNA structure will be a separate entry (Fig. 1b). Then, the spatial-temporal features of sgRNA sequence and structure features will be extracted through convolutional recurrent neural networks (CRNN) separately. CRNN is the combination of two of the most prominent neural networks (convolutional neural network and recurrent neural network), which is designed to extract features in both spatial and temporal dimensions. It starts with a traditional 2D convolutional neural network followed by batch normalization and RELU activation. Two such convolution layers are placed in a sequential manner with their corresponding activations. The convolutional layers are then also followed by dropout (with a dropout rate of 50%) and max pooling. Next, a Long short-term memory (LSTM) layer is applied for temporal feature extraction followed by dropout (with a dropout rate of 30%) and the dense layer. Finally, the spatial-temporal features from sequence and structure are concatenated in one single layer followed by fully connected neural network for the prediction. A Deep Score for each sgRNA will be output to indicate the on-target efficiency of a specific sgRNA. The higher the Deep Score is, the more likely the sgRNA is to be effective.

Hyperparameter Tuning was performed by HParams Dashboard in TensorBoard, which provides several tools to identify the most promising sets of hyperparameters. Here, we focused on four parameters: the number of units in the dense layer, the dropout rate, the optimizers, and the learning rate. For simplicity, each parameter was treated as a discrete value and a grid search was used to try all combinations of these discrete parameters. The Mean Squared Error (MSE) of the model at every epoch was tracked to determine the best combination of hyperparameters.

## Performance comparison of DeepCas13 with conventional machine learning

Five conventional machine learning methods were used in this comparison, including Random Forest, XGBoost, Support Vector Machine, AdaBoost, and Gradient Boosting. All of the methods were implemented using scikit-learn package[59] except XGBoost. A total of 185 features were extracted for the training based on previous study[18]. The training data contained 5726 sgRNAs from three Cas13d tiling screening experiments (*CD46*, *CD55*, *CD71*). Among them, 1174 sgRNA were marked as positive samples as their LFC < = −0.5. We performed 5-fold cross validation to evaluate model performance on these limited training samples. To make the comparison more comprehensive, four independent indicators were used to measure the predictive performance, including R squared score, Pearson correlation coefficient, area under the ROC curve, and area under the Precision-Recall curve.

## Performance comparison of DeepCas13 with start-of-the-art tool

Two Cas13d pooled proliferation screening datasets (performed on A375 cells) were used as evaluated data in this comparison separately. For each target gene, only the top 4 or bottom 4 ranked sgRNAs were left for further analysis. To increase the training set, the Cas13d tiling screening data was also combined with the training data. We also performed five-fold cross validation for calculating Deep Scores. Both tiling screening and proliferation datasets are used for training and testing. After five-fold cross validation, only these sgRNAs from proliferation dataset are left for further evaluation. The $RF_{NBT}$ scores were downloaded from cas13design Git archive directly for public A375

dataset. For A375 data from our experiment, the $RF_{NBT}$ scores were calculated by its source code. Due to built-in filtering setting in $RF_{NBT}$, some guides in our library cannot get a $RF_{NBT}$ scores. So, only these sgRNAs with both Deep Score and $RF_{NBT}$ score were left for further comparison. We compared the performance between DeepCas13 and $RF_{NBT}$ by Pearson correlation coefficient, AUC score and AUPR score.

For leave-one-dataset-out evaluation, DeepCas13 was trained on three tiling screening datasets and one A375 pooled screening dataset, and another A375 pooled screening data was used as an independent validation dataset. We also compared the performance when trained DeepCas13 only using FACS sorting data. The AUPR score and AUC score were used to evaluate the performance.

## Performance evaluation on circRNA screening

For dataset from circRNA screening study[42], we performed five-fold cross validation to get Deep Scores for these limited training sgRNAs. Any sgRNAs with LFC < = −0.5 were set as positive samples and other sgRNAs were set as negative samples. We first calculated the AUC score and AUPR score for the whole dataset. Next, the top 10, 15, 20, 25, and 30% negative selected circRNAs were chosen from rank list in the gene summary table separately. Only these sgRNAs that associated to the top negative selected circRNAs were left for calculating the AUPR score.

For dataset from another circRNA screening study[43], top 50 negative selected circRNAs were selected from the gene summary table for HEK293FT, HT29, and Hela cell lines separately. The sgRNAs associated to these top negative selected circRNAs were used to generate Precision-Recall curve and calculate AUPR score. We also calculated the AUC scores and AUPR scores for all the sgRNAs in the library.

## Off-target viability analysis

The random forest method was implemented using scikit-learn package[59]. All guides targeting non-essential genes in two studies are used (499 guides targeting 10 non-essential genes in our study, and 2593 guides targeting 65 non-essential genes in research[18], respectively). We compare guides that have the strongest dropout (lowest 25% log fold change value) with the rest of the guides.

## lncRNA expression and survival analysis

The lncRNA expressions from A375 and A549 cells are from the Cancer Cell Line Encyclopedia (CCLE) and are downloaded from DepMap (version: 2019)[60]. The survival analysis of lncRNAs is performed using "The Atlas of Noncoding RNAs in Cancer" (TANRIC) database[61] and DrBioRight platform[62].

## Statistics and reproducibility

The number of sample size (*n* number) and statistical details are indicated in the respective figure legends. All values for n are for individual samples or individual experiments. Sample sizes were chosen at least three to be the minimum for any inferential analysis. No data were excluded from the analyses. The two-sided independent t-test is used for calculating the *p*-value. The top, mid-line, and bottom of the boxplot represents the upper quartile (Q3), median, and lower quartile (Q1), respectively. The ends of the whiskers represent the minimum and maximum values in the data set. The 5-fold cross−validation in scikit-learn package is used to test the ability of machine learning models to predict sgRNA efficiency.

## Reporting summary

Further information on research design is available in the Nature Portfolio Reporting Summary linked to this article.

# Data availability

Both primary and secondary raw Cas13d screening data generated in this study have been deposited in Gene Expression Omnibus (GEO)

under the accession number GSE183256. Source data for the figures are provided with this paper. Source data are provided with this paper.

## Code availability

The DeepCas13 website can be accessed at http://deepcas13.weililab. org and the standalone program is available on BitBucket https:// bitbucket.org/weililab/deepcas13/src/master/.

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

## Acknowledgements

We would like to thank all members of the Li and Fei lab for the valuable discussion of the results. This work was supported by the startup fund from the Center of Genetic Medicine Research and Gilbert Family NF1 Institute at the Children's National Medical Center (X.C. and W.L.), the National Institute of Health research grant R01 HG010753 (W.L.), the National Natural Science Foundation of China 31871344 and 32071441 (T.F.), the Fundamental Research Funds for the Central Universities N182005005 and N2020001 (T.F.), the 111 Project B16009 (T.F.), and LiaoNing Revitalization Talents Program XLYC1807212 (T.F.), the Key Laboratory of Bioresource Research and Development of Liaoning Province (2022JH13/10200026) (T.F.).

## Author contributions

W.L. and T.F. conceived and designed the study. W.L., X.C., and Z.L. designed the library. Z.L., T.F., W.Z., and H.Z. performed Cas13d screens. Zihan L. and S.W. contributed to other experiments. X.C., W.L., and Z.L. processed, analyzed, and interpreted the data. X.C. and R.S. designed and implemented the DeepCas13 website with the help of J.P. X.C., W.L., and T.F. wrote the manuscript with the help of other authors. W.L. and T.F. supervised the study. All authors read and approved the final manuscript.

## Competing interests

W.L. is a paid consultant to Tavros Therapeutics, Inc. The other authors declare no competing interests.
