## [Peer Review File · Nature Communications]

Reviewers' Comments:

Reviewer #1:

Remarks to the Author:

In the submission, the authors have proposed an approach to model the on-target and off-target effects of CRISPR-Cas13d. The topic is trendy and well-motivated in recent years. To avoid side-effect of genomic editing systems, it is always desired to have the proposed approach to address it. Technically, the authors have involved the designs of deep learning which is the state-of-the-arts approach in the recent surge of AI. For data, the authors have also generated their own data which was combined with the other data as a relatively comprehensive dataset. The results indicate that the proposed approach (DeepCas13) demonstrated its good performance. Overall, the manuscript is well-organized and nicely written. Therefore, I only have few comments.

1. The authors have combined their data with other existing data to form a good benchmark dataset. Nonetheless, I was wondering if there is any batch effect or cross-platform or cross lab biases there. Some comparison results would be interesting.
2. It is very nice for me to follow the design of the proposed approach. From the computational methodology, I have no further comments on it. From the biological point of view, it could be nice if the authors can elucidate the motifs as captured within the deep learning models since those motifs can be related to the targeting efficiency.
3. A website has been built as given at the end of the abstract. To attract citations, it would also be valuable if the authors can release the corresponding standalone programs on the website for high-throughput command line usage.
4. There are other methods for other CRISPR systems. It would be best if the authors can briefly discuss it in the literature review and make the high-level comparisons if applicable.

Reviewer #2:

Remarks to the Author:

This paper proposed an interesting work. The authors performed CRISPR-Cas13d proliferation screens that target protein-coding genes and long non-coding RNAs (lncRNAs), followed by a systematic modeling of Cas13d on-target efficiency and off-target viability effect. Then they first designed a deep learning model, named DeepCas13, to predict the on-target activity of a gRNA with high accuracy from its sequence and secondary structure. DeepCas13 outperforms existing methods and accurately predicts the efficiency of guides targeting both protein-coding and non-coding RNAs. Next, they systematically studied guides targeting non-essential genes, and found that the off-target viability effect is closely related to their on-target RNA cleavage efficiency. Finally, they applied these models to their screens that included guides targeting 234 lncRNAs, and identified lncRNAs that affect cell viability and proliferation in multiple cell lines. The work is explained in a clear manner in the paper.

However, there are several problems that need to be explained and improved.

1. -lines 88 and 131 both mentioned that "We obtained data from 22,599 Cas13d sgRNAs to systematically investigate the efficiency and specificity of Cas13d". Are all the 22,599 Cas13d sgRNAs used for training the Final DeepCas13 model? If so, we suggest the authors to provide the dataset as a Supplementary File.

2. -Line 149: "We compared DeepCas13 with five conventional machine learning methods". If the DeepCas13 model trained and tested on the same datasets with the five conventional machine learning methods?

-Line 171: "We next compared DeepCas13 with a recently published random forest prediction model¹⁸".

When compared with RFNBT, if the authors used the same training set as deepcas13 model to retrain the RF model or just only used FACS screening datasets for training RFNBT?

3. In the construction of DeepCas13, the process of selecting and determining hyperparameters is not described in detail. We suggest that the authors should describe the value of the hyperparameters in detail or provide some key code for model building so that the reader can

reproduce the model.

4. -Line159: "LFC", the abbreviation of "log fold change", is recommended to be placed right behind the first occurrence.

-Line194 and 196: We suggest that the author should change "AUROC" into "AUC".

Reviewer #3:

Remarks to the Author:

Cheng et al developed a deep learning model to predict Cas13d gRNA on-target efficiency, taking into account gRNA sequence and secondary structure. The authors showed that the model outperforms existing methods in predicting Cas13d gRNA efficiency for both protein-coding and non-coding RNAs including circRNAs and lncRNAs. The authors also found that gRNAs targeting non-essential genes confer unintended effects on cell viability that are similar to on-target RNA cleavage efficiency. The authors proposed that gRNAs targeting non-essential genes should be used as negative controls in proliferation screens to reduce false positives.

The study used a comprehensive list of publicly available and in-house Cas13d screening datasets to train the model. The model has the power to predict efficient on-target gRNAs for not just protein-coding RNAs but also noncoding RNAs. Overall, the findings are interesting, and the DeepCas13 model can be useful for the scientific community. However, additional evidence is needed to demonstrate that the model is truly predictive of efficient Cas13d gRNAs for both protein-coding and non-coding RNAs.

Major:

1. DeepCas13 uses predicted secondary structure information as input for training. RNA structure inside cells is quite complex due to the involvement of RNA-protein interactions and RNA localization. Minimum free energy (MFE) from RNA secondary structures predicted by ViennaRNA likely does not account for those events inside cells and may not accurately predict RNA folding. Can information from SHAPE (RNA structure analysis by selective 2'-hydroxyl acylation analyzer by primer extension) reactivity datasets improve the model better than MFE?

Noncoding RNAs in particular interact with multiple RNA-binding proteins to function inside cells, and MFE can be a poor predictor of how the RNAs fold as well as accessible regions for gRNA targeting.

2. RNAs that are low abundant are usually more difficult to be depleted, particularly non-coding RNAs, which are typically expressed at lower levels than protein-coding RNAs. In a CRISPR screen library, even efficient gRNAs may not be able to deplete those low-abundant RNA targets, and the gRNAs may be mistakenly considered inefficient. Does DeepCas13 account for this issue when using the screening data for training?

3. To demonstrate that DeepCas13 is truly predictive of efficient gRNAs and is superior to other algorithms, such as the one developed by Wessels et al (PMID: 32518401). It would be convincing to experimentally compare head-to-head the top gRNAs (e.g. top 10 gRNAs) predicted by each algorithm for a set of different RNA targets/genes (e.g. >10 mRNAs, >10 lncRNAs and >10 circRNAs). The on-target efficiency can be measured as how well the RNA target abundance gets depleted by the predicted gRNAs or the number of predicted gRNAs that achieve certain levels of RNA depletion. Other experimental approaches to evaluate the model's predictive power and benchmark against existing algorithms beyond using published data for comparison would provide more confidence in the predictive strength and value of DeepCas13.

Minor:

1. It would be clearer if a table listing all datasets that were used to train the model is provided.

Summary

We appreciate the insightful comments of the three anonymous reviewers and revised the manuscript according to their reviews. Specifically, we made the following major changes:

- We performed a secondary CRISPR-Cas13d screen to systematically evaluate the performance of DeepCas13 and another method (RF_{NBT}). The screening approach tested the guides at a much larger scale (over 9,000 guides in one single experiment). The results confirmed the higher prediction accuracy of DeepCas13 and are now presented in the new Figure 6;
- In addition to screens, we performed qRT-PCR validation experiments on the top predicted guides (Fig. 6), including a “head-to-head” comparison (recommended by the reviewer) which directly evaluate the performances of DeepCas13 and RF_{NBT};
- We applied Integrated Gradient (IG) analysis and identified sequence motifs that determine guide efficiency;
- We provided a stand-alone program for users to run batch predictions; and
- Other revisions recommended by the reviewers.

Changes in the manuscript are **highlighted** within the manuscript and a point-by-point response to the reviewers' comments are provided below.

We apologize for the delay in revising this manuscript, due to the multiple lockdowns in several major cities in China, which severely delayed our experimental validations.

Reviewers' Comments to the Authors:

Referee: 1

1. The authors have combined their data with other existing data to form a good benchmark dataset. Nonetheless, I was wondering if there is any batch effect or cross-platform or cross lab biases there. Some comparison results would be interesting.

Response: We argue that there are few, if not none, batch effects in our study. This is because batch effects are usually detected and removed at the raw assay signals (e.g., gene expressions in RNA-seq). In contrast, we calculated LFC values between treatment vs. control samples within each dataset, and combined the LFC values from multiple datasets together. Because we use the corresponding control samples within each dataset, the batch effect that is specific to each dataset is minimized. In fact, we combined two types of Cas13d screens, including proliferation screens and FACS-based screens (targeting CD71, CD55, CD46). We checked the LFC distributions of both screening types, and found similar LFC distributions, especially for guide RNAs that are efficient (LFC < -

0.5; Reviewer-Only-Figure1). Therefore, few or no batch effects exist in our combined datasets.

Reviewer-Only-Figure 1. Guide LFC distribution

2. It is very nice for me to follow the design of the proposed approach. From the computational methodology, I have no further comments on it. From the biological point of view, it could be nice if the authors can elucidate the motifs as captured within the deep learning models since those motifs can be related to the targeting efficiency.

Response: We applied Integrated Gradients (IG; (Sundararajan, Taly, and Yan 2017)), one commonly used explainable AI technique, to calculate motif or base preference from the guide sequence branch in our deep learning model. We applied IG to capture the motif/base preference of the extended guide sequence (± 5 bp) in our DeepCas13 model. As shown in Reviewer-Only-Figure2 and the new Fig. 2g and Fig. S5e-f, we found highly efficient guides prefer C/G rather than A/T in positions 14-22nt. The GC-rich motif in this position is consistent with the findings in another bioRxiv preprint (Wei et al. 2021).

Reviewer-Only-Figure 2 (also in the revised Fig. 2g). Base preference of Cas13d guide RNA based on IG

3. A website has been built as given at the end of the abstract. To attract citations, it would also be valuable if the authors can release the corresponding standalone programs on the website for high-throughput command line usage.

Response: As suggested by the reviewer, we created a standalone program and deposited our source code on BitBucket (<https://bitbucket.org/weililab/deepcas13/>). We also provided documentations and demos on installation, training, prediction and designing. In Data Availability section, we added the sentence that “[the standalone program is available on BitBucket https://bitbucket.org/weililab/deepcas13/](https://bitbucket.org/weililab/deepcas13/)”.

4. There are other methods for other CRISPR systems. It would be best if the authors can briefly discuss it in the literature review and make the high-level comparisons if applicable.

Response: We added a brief discussion in the Discussion (Line 415-420) and Introduction sections (Line 59-68), we briefly compared DeepCas13 with several deep learning (DL)-based tools designed for CRISPR Cas9 system. The main difference between our tool and other DL tools is that, DeepCas13 uses features from both sgRNA sequence and RNA secondary structure, while other DL tools only use sequence features. The secondary structure of sgRNAs can improve the DL performance, as is shown in Fig. S2b.

Referee: 2

1. -lines 88 and 131 both mentioned that “We obtained data from 22,599 Cas13d sgRNAs to systematically investigate the efficiency and specificity of Cas13d”. Are all the 22,599 Cas13d sgRNAs used for training the Final DeepCas13 model? If so, we suggest the authors to provide the dataset as a Supplementary File.

Response: These 22,599 sgRNAs target protein-coding genes, lncRNAs and circRNAs. Among those, 10,279 of guides that target essential genes (in proliferation screens) or marker genes (in FACS screens) were used for training. We clarified this number in the revision (Line 140), and added these 10,279 guides to the Supplementary File 1 with guide sequence and LFC info.

2. -Line 149: “We compared DeepCas13 with five conventional machine learning methods”. If the DeepCas13 model trained and tested on the same datasets with the five conventional machine learning methods? -Line 171: “We next compared DeepCas13 with a recently published random forest prediction model¹⁸”. When compared with RF_{NBT}, if the authors used the same training set as deepcas13 model to retrain the RF model or just only used FACS screening datasets for training RF_{NBT}?

Response: For the first question: yes, in order to make a fair comparison, all methods (including DeepCas13) were trained and tested on the same datasets.

For the second question: we used the existing RF_{NBT} model that is trained by the authors of the NBT paper (Wessels et al. 2020). We did not retrain the RF model because (1) we

already included a similar random forest (RF) method during the comparison, which is fair because all methods are trained and tested on the same datasets; (2) the training procedure of RF_{NBT} is complex and it includes several ad-hoc preprocessing and postprocessing steps that may affect the final performance. Therefore, we decided to use existing RF_{NBT} model and did not retrain it. We clarified this point in the revised manuscript (Line 181).

3. *In the construction of DeepCas13, the process of selecting and determining hyperparameters is not described in detail. We suggest that the authors should describe the value of the hyperparameters in detail or provide some key code for model building so that the reader can reproduce the model.*

Response: Hyperparameter Tuning was performed by HParams Dashboard in TensorBoard, which provides several tools to identify the most promising sets of hyperparameters. Here, we focused on four parameters: the number of units in the dense layer, the dropout rate, the optimizers and the learning rate. For simplicity, each parameter was treated as a discrete value and a grid search was used to try all combinations of these discrete parameters. The Mean Squared Error (MSE) of the model at every epoch was tracked to determine the best combination of hyperparameters. The final hyperparameters and pre-trained model can be found at our public repository on BitBucket (<https://bitbucket.org/weillilab/deepcas13/src/master/>). We clarified the Hyperparameter Tuning process in the revised methods section.

4. *-Line159: “LFC”, the abbreviation of “log fold change”, is recommended to be placed right behind the first occurrence. -Line194 and 196: We suggest that the author should change “AUROC” into “AUC”.*

Response: We revised the manuscript accordingly.

Referee: 3

Major:

1. *DeepCas13 uses predicted secondary structure information as input for training. RNA structure inside cells is quite complex due to the involvement of RNA-protein interactions and RNA localization. Minimum free energy (MFE) from RNA secondary structures predicted by ViennaRNA likely does not account for those events inside cells and may not accurately predict RNA folding. Can information from SHAPE (RNA structure analysis by selective 2'-hydroxyl acylation analyzer by primer extension) reactivity datasets improve the model better than MFE?*

Noncoding RNAs in particular interact with multiple RNA-binding proteins to function inside cells, and MFE can be a poor predictor of how the RNAs fold as well as accessible regions for gRNA targeting.

Response: We used a graph-kernel-based machine learning approach, ShaKer (Mautner et al. 2019), to predict the SHAPE reactivity information from the direct repeats and gRNA sequence on the FACS tiling dataset (5,726 guides). To examine whether SHAPE prediction is better than MFE, we revised our deep learning model such that the predicted SHAPE reactivity vector, instead of MFE vector, were used as the input of the Fold branch in DeepCas13. Compared to MFE, SHAPE have similar performances (t-test p -value for AUC, Pearson Correlation and AUPR is 0.8333, 0.8519 and 0.6788 respectively), as are shown in Reviewer-Only-Figure3. Besides the similar performance, the average running time to calculate MFE by ViennaRNA is much shorter than calculating SHAPE reactivity by ShaKer (0.002s for ViennaRNA vs. 4.744s for ShaKer). The comparison was performed on a HPC cluster (CPU: Xeon-E5-2630 2.20GH, Memory: 512GB). For these reasons, we use MFE as input of our machine learning model.

It is important to note that in order for a prediction tool to be generally applicable, the ShaKer prediction algorithm, rather than experimentally generated SHAPE information, was used. Therefore, it is still unclear whether MFE or SHAPE is better, as the performance of SHAPE prediction algorithms (like ShaKer) will also be critical. This would be an interesting and important question to be investigated in the future.

Reviewer-Only-Figure 3. Performance between MFE and SHAPE

2. RNAs that are low abundant are usually more difficult to be depleted, particularly non-coding RNAs, which are typically expressed at lower levels than protein-coding RNAs. In a CRISPR screen library, even efficient gRNAs may not be able to deplete those low-abundant RNA targets, and the gRNAs may be mistakenly considered inefficient. Does DeepCas13 account for this issue when using the screening data for training?

Response: DeepCas13 was trained on guides only targeting essential genes, which are usually highly expressed in multiple cell lines including A375. In fact, their expression distribution is much higher than all the other genes from the A375 expression profiles of DepMap (Reviewer-Only-Figure 4a). Though the expression of these essential genes are not in the same level (range from 3.30 to 11.96), there are no significant difference of the sgRNA LFC distribution (Reviewer-Only-Figure 4b, $p=0.2975$). Therefore, we did not account for gene expression in DeepCas13, because all the genes are highly expressed.

For non-coding RNAs, their expressions are usually much lower than essential genes. For the exact reason mentioned by the reviewer, we did not use guides targeting non-coding RNAs for training. We clarified this in the revised manuscript (Line 140).

Reviewer-Only Figure 4. a. The expressions of all genes (black) and essential genes used for DeepCas13 training. The gene expressions of A375 cells in DepMap were used. **b.** sgRNA LFC distribution at different gene expression manner.

3. To demonstrate that DeepCas13 is truly predictive of efficient gRNAs and is superior to other algorithms, such as the one developed by (Wessels et al. 2020). It would be convincing to experimentally compare head-to-head the top gRNAs (e.g. top 10 gRNAs) predicted by each algorithm for a set of different RNA targets/genes (e.g. >10 mRNAs, >10 lncRNAs and >10 circRNAs). The on-target efficiency can be measured as how well the RNA target abundance gets depleted by the predicted gRNAs or the number of predicted gRNAs that achieve certain levels of RNA depletion. Other experimental approaches to evaluate the model's predictive power and benchmark against existing algorithms beyond using published data for comparison would provide more confidence in the predictive strength and value of DeepCas13.

Response: We thank the reviewer's critical comment, which enabled us to rigorously evaluate the performance of DeepCas13 using experimental approaches in the revision. Because a large number of qRT-PCR experiments is needed for head-to-head comparison (10 gRNAs * 2 methods * 30 mRNA/lncRNA/circRNA * 3 replicates = 1,800 individual experiments), we decided to perform a new round of Cas13d screen, together with qRT-PCR validation on a few gRNAs. The results are now presented in the new Figure 6 and Figure S8, and in Reviewer-Only-Figure 5 below. Note that the screening approach enabled us to systematically compare two methods at a much larger scale, as the screening tested the performances of 9,500 gRNAs targeting 59 mRNAs, 38 lncRNAs and 40 circRNAs. As these targets are reported to be essential in general or specific cell lines, the efficiency of their knockdown can be indirectly measured by cell viability-based Cas13d screening in a high-throughput manner (Reviewer-Only-Figure 5b).

Reviewer-Only-Figure 5 (also Fig. 6 in revised manuscript). DeepCas13 performance validation.

Both screening and qRT-PCR data demonstrated the higher prediction accuracy of DeepCas13 than the existing method (RF_{NBT}). For example, the top 10% efficient sgRNAs predicted by DeepCas13 have a significantly lower LFC distribution than those predicted by RF_{NBT} (Wessels et al. 2020), in both mRNA and lncRNA groups (Reviewer-Only-Figure 5b). For circRNAs, DeepCas13 is slightly better than RF_{NBT} , although the difference is not significant. The better performance of DeepCas13 is also confirmed at the gene level (Reviewer-Only-Figure 5c), where we selected the top 10 (marked as H) and bottom 10 predicted efficient sgRNAs (marked as L). The top sgRNAs predicted by DeepCas13 only,

but not $RF_{NBT}(H_{DeepCas13})$ have a much lower LFC than the top sgRNAs predicted by RF_{NBT} , but not DeepCas13.

For qRT-PCR validation, we performed two sets of experiments. First, we selected 10 sgRNAs in the screening library (4 for mRNA and 6 for circRNA) that are predicted to be highly efficient by each method, and measured the reduction of its target abundances (Reviewer-Only-Figure 5e). sgRNAs predicted by DeepCas13 reduced target expression more efficiently than those predicted by RF_{NBT} . Next, we performed head-to-head comparison, as recommended by the reviewer, by designing 18 new sgRNAs targeting 9 different genes, including 9 sgRNAs with high Deep Score and 9 sgRNAs with high RF_{NBT} score. The guides are selected such that they did not overlap with the existing guides in our first and secondary screen. In five out of the six targets (Reviewer-Only-Figure 5f), DeepCas13 showed better performance than RF_{NBT} , including 3 statistically significant targets. Note that three targets (1 gene, two lncRNAs) were excluded because we observed target activation effect rather than inhibition effect (Fig. S8g). In conclusion, DeepCas13 demonstrated better performance in identifying efficient sgRNAs than the existing tool (Reviewer-Only-Figure 5).

Minor:

1. It would be clearer if a table listing all datasets that were used to train the model is provided.

Response: All the 10,279 guides used to train the DeepCas13 model have been provided in the Supplementary File 1 with both guide sequence and LFC info.

References

- Mautner, Stefan, Soheila Montaseri, Milad Miladi, Martin Raden, Fabrizio Costa, and Rolf Backofen. 2019. "ShaKer: RNA SHAPE Prediction Using Graph Kernel." *Bioinformatics (Oxford, England)* 35 (14): i354–59.
- Sundararajan, Mukund, Ankur Taly, and Qiqi Yan. 2017. "Axiomatic Attribution for Deep Networks." *ArXiv [Cs.LG]*. arXiv. <http://arxiv.org/abs/1703.01365>.
- Wei, Jingyi, Peter Lotfy, Kian Faizi, Eleanor Wang, Hannah Slabodkin, Emily Kinnaman, Sita Chandrasekaran, et al. 2021. "Deep Learning and CRISPR-Cas13d Ortholog Discovery for Optimized RNA Targeting." *BioRxiv*. <https://doi.org/10.1101/2021.09.14.460134>.
- Wessels, Hans-Hermann, Alejandro Méndez-Mancilla, Xinyi Guo, Mateusz Legut, Zharko Daniloski, and Neville E. Sanjana. 2020. "Massively Parallel Cas13 Screens Reveal Principles for Guide RNA Design." *Nature Biotechnology* 38 (6): 722–27.

Reviewers' Comments:

Reviewer #1:

Remarks to the Author:

The authors have addressed my comments.

Reviewer #2:

Remarks to the Author:

Thank you for your considering our comments, you have made further improvements to the manuscript and I think it reaches the level can be published.

Reviewer #3:

Remarks to the Author:

The authors addressed all of my concerns well. I appreciate that the authors went the extra mile to perform an unbiased CRISPR screen for benchmarking the machine learning approach against an existing algorithm. The additional figures certainly strengthened the manuscript. Congratulations. Well deserved.

Manuscript ID: NCOMMS-21-36492

Response to Reviewers

Reviewers' Comments to the Authors:

Referee: 1

The authors have addressed my comments.

Referee: 2

Thank you for your considering our comments, you have made further improvements to the manuscript and I think it reaches the level can be published.

Referee: 3

The authors addressed all of my concerns well. I appreciate that the authors went the extra mile to perform an unbiased CRISPR screen for benchmarking the machine learning approach against an existing algorithm. The additional figures certainly strengthened the manuscript. Congratulations. Well deserved.

Response: We thank the reviewers for their positive comments.